# Parasitization of *Aphis gossypii* Glover by *Binodoxys communis* Gahan Causes Shifts in the Ovarian Bacterial Microbiota

**DOI:** 10.3390/insects14040314

**Published:** 2023-03-25

**Authors:** Jinming Li, Zhe An, Junyu Luo, Xiangzhen Zhu, Li Wang, Kaixin Zhang, Dongyang Li, Jichao Ji, Lin Niu, Xueke Gao, Jinjie Cui

**Affiliations:** 1Zhengzhou Research Base, State Key Laboratory of Cotton Biology, School of Agricultural Sciences, Zhengzhou University, Zhengzhou 450001, China; 2State Key Laboratory of Cotton Biology, Institute of Cotton Research, Chinese Academy of Agricultural Sciences, Anyang 455000, China

**Keywords:** parasitoids, *Buchnera*, *Serratia*, *Arsenophonus*, host–parasite interactions, aphid, agricultural pest control

## Abstract

**Simple Summary:**

*Aphis gossypii* Glover is an important agricultural pest distributed worldwide, which can reduce the yield of cotton crops and cause huge economic losses. *Binodoxys communis* Gahan is the main parasitoid wasp of *A. gossypii*. Previous studies have shown that parasitization causes reduced egg production in *A. gossypii*, but the effects on the symbiotic bacteria of the host ovaries are unknown. In this study, we analyzed the microbial communities in cotton aphid ovaries by 16S rDNA sequencing and their changes before and after parasitization, performed a functional prediction analysis of the microbial communities in cotton aphid ovaries, and finally performed RT-qPCR on some core symbiotic bacteria. In summary, our results provide a framework for investigating shifts in the microbial communities in host ovaries and broaden our understanding of the interactions among aphids, parasitoid wasps, and endosymbionts.

**Abstract:**

Background: *Aphis gossypii* Glover is an important agricultural pest distributed worldwide. *Binodoxys communis* Gahan is the main parasitoid wasp of *A. gossypii*. Previous studies have shown that parasitization causes reduced egg production in *A. gossypii*, but the effects of parasitism on the symbiotic bacteria in the host ovaries are unknown. Results: In this study, we analyzed the microbial communities in the ovaries of *A. gossypii* without and after parasitization. Whether parasitized or not, *Buchnera* was the dominant genus of symbiotic bacteria in the ovaries, followed by facultative symbionts including *Arsenophonus*, *Pseudomonas*, and *Acinetobacter*. The relative abundance of *Buchnera* in the aphid ovary increased after parasitization for 1 d in both third-instar nymph and adult stages, but decreased after parasitization for 3 d. The shifts in the relative abundance of *Arsenophonus* in both stages were the same as those observed for *Buchnera.* In addition, the relative abundance of *Serratia* remarkably decreased after parasitization for 1 d and increased after parasitization for 3 d. A functional predictive analysis of the control and parasitized ovary microbiomes revealed that pathways primarily enriched in parasitization were “amino acid transport and metabolism” and “energy production and conversion.” Finally, RT-qPCR analysis was performed on *Buchnera*, *Arsenophonus*, and *Serratia*. The results of RT-qPCR were the same as the results of 16S rDNA sequencing. Conclusions: These results provide a framework for investigating shifts in the microbial communities in host ovaries, which may be responsible for reduced egg production in aphids. These findings also broaden our understanding of the interactions among aphids, parasitoid wasps, and endosymbionts.

## 1. Introduction

The evolution of insects and their bacterial symbionts is ubiquitous in nature. Symbiotic bacteria associated with insect hosts can have different biological effects on their hosts [1,2,3]. As a source of metabolism for their insect hosts, some of these symbiotic bacteria are essential for the survival and reproduction of their hosts and are considered obligate symbionts. In contrast, others are facultative symbionts, which are non-essential for the growth or development of their hosts but can provide other benefits to their host such as defense against pathogens and natural enemies [4,5], body color regulation [6], heat tolerance [7], and modulation of host reproduction [8]. However, a growing number of studies have shown that facultative symbiotic bacteria also play a key role in interfering with host–parasite interactions [9]. A variety of insect microbes of different microbial lineages have been shown to have important protective effects against fungal pathogens [10,11], viruses [12], predators [13], parasitoids [14,15], and parasitoid nematodes [16].

Aphids are remarkable models of host–microbe evolution as they contain a wide range of facultative symbiotic bacteria in addition to their ancient obligate symbiotic bacteria *Buchnera aphidicola*, all of which are transmitted vertically from the mother to the offspring in their bodies [17,18]. Currently, the cotton aphid (*Aphis gossypii* Glover) is a global agricultural pest that causes damages in a wide variety of plants by feeding on their bast [19]. Cotton aphids are widely distributed and can breed on more than 600 species of plants in tropical, subtropical, and temperate regions. *A. gossypii* contains the obligate endosymbiont *Buchnera aphidicola*, which provides essential amino acids for aphids. There are also several common facultative symbiotic bacteria, such as *Arsenophonus*, *Serratia*, and *Hamiltonella. Serratia symbiotica* and *Hamiltonella defense* protect aphids against heat shock [20] and provide resistance to parasitoid wasps [21]. *Regiella insecticola* is able to resist fungi [11] and parasitoids and enhance host plant fitness [22]. *Rickettsiella* can change the host body color [6]. At this stage, the control of aphids relies primarily on chemical pesticides, but many chemical pesticides are not target-specific and are toxic to a wide range of organisms [23]. In contrast, biological control using natural enemies is not only safe and non-polluting, but also sustainable. Parasitoid wasps are an important mechanism of biological control and are characterized by high reproductive power, short life cycle, rapid spread throughout the crop, and specific targeting [24]. 

*Binodoxys communis* Gahan (Hymenoptera:Braconidae) is an endoparasitoid wasp that is known to attack several Aphids species, in particular soybean aphid, *Aphis glycines* (Matsumura), and melon aphid, *A. gossypii* [25]. Female parasitoid wasps use their ovipositors to lay eggs inside aphids, and parasitoid wasp eventually cause the death of the host by consuming the living host’s developing tissue [26]. When cotton aphids are attacked by parasitoid wasps, the larvae of the parasitoid wasps can alter the host–symbiont dynamics and consume host-provided nutrients [27]. Furthermore, our previous experiments showed that *Lysiphlebia japonica* was able to cause a significant reduction in *A. gossypii* fecundity through parasitism [28]. However, the mechanism of how parasitoid wasps modulate aphid ovary symbiont communities remains unknown.

Although changes in bacterial communities due to parasitism have been widely studied in pea aphids [29], little is known about the effects of parasitism on symbiotic bacteria in *A. gossypii*. As parasitism decreases *A. gossypii* fecundity [28], it is crucial to understand the changes in symbiotic bacteria within the cotton aphid ovaries. Therefore, in order to better understand the symbiotic bacteria changes in the cotton aphid ovaries after parasitization, we conducted high-throughput 16S rDNA sequencing to analyze the ovary microbiome of parasitized and non-parasitized cotton aphids. Specifically, changes in the endosymbiotic bacterial community in the ovaries of parasitized cotton aphid nymphs and adults were examined at 1 d and 3 d; insects not parasitized during the same period served as controls. Through this characterization and analysis of the changes of the ovary bacterial communities, we aimed to understand the composition of symbiotic bacteria within the ovaries of cotton aphids at different developmental stages and how parasitism affects the symbiotic bacteria community in cotton aphid ovaries.

## 2. Materials and Methods

### 2.1. Insect Collection

The *A. gossypii* and *B. communis* used in this experiment were from the cotton field of the Cotton Research Institute of the Chinese Academy of Agricultural Sciences (36°5′34.8″ N, 114°31′47.19″ E). The laboratory feeding conditions of *A. gossypii* were as follows: fed cotton leaves, 26 ± 1 °C, 65 ± 5% humidity, and a 16:8 h light/dark cycle. *B*. *communis* were reared in cotton aphids kept at 24 ± 1 °C, 75 ± 5% humidity, and a 14:10 h light/dark cycle.

### 2.2. Sample Collection and Processing

In this experiment, we studied parasitized and unparasitized cotton aphids in two development periods: third-instar nymph and adult. In each development period, each experimental group consisted of sixty aphids: thirty parasitized and thirty unparasitized, and there were six replicates of each experimental group. To collect parasitized aphids, the aphids were exposed to parasitoid wasps until parasitism was observed, and then the aphids were removed. In order to obtain high-quality data and minimize the influence of parasitoid wasp eggs on the results, the parasitoid wasp eggs in parasitized aphids were removed via dissection under a microscope, followed by whole-surface sterilization of the cotton aphid ovaries before subsequent analysis. To remove microbial contaminants from the insect surfaces prior to PCR amplification and sequencing, each sample was soaked in 70% ethanol for 5 min, bleached at 10% for 30 s, and then washed with sterile ultrapure water.

### 2.3. DNA Extraction, PCR Amplification, Library Preparation, and Sequencing

Total genomic DNA from the four different groups was extracted using the E.Z.N.A.^®^ soil DNA kit (Omega Bio-tek, Norcross, GA, USA) according to the manufacturer’s instructions. DNA quality was assessed by 1% agarose gel electrophoresis, and DNA concentration and purity were determined with NanoDrop 2000. A nearly 420 bp V3–V4 region of the 16S rRNA gene was amplified with the primers 338F (5′-ACTCCTACGGGAGGCAGCAG-3′) and 806F (5′-GGACTACHVGGGTWTCTAAT-3′). The PCR reaction system contained 4 μL of 5× FastPfu Buffer, 0.4 μL of FastPfu DNA polymerase (Transgene, Beijing, China), 2 μL of dNTPs, 0.8 μL of forward and reverse primers (μM), and 10 ng of DNA. The PCR amplification cycle conditions were: 95 °C for 3 min, 27 cycles of 95 °C for 30 s, 53 °C for 30 s, 72 °C for 45 s, followed by a final elongation step at 72 °C for 10 min and storing at 4 °C. The PCR product was recovered using a 2% agarose gel. The AxyPrep DNA Gel Extraction Kit (Axygen Biosciences, Union City, CA, USA) was used for purification, and the Quantus™ Fluorometer (Promega, Madison, WI, USA) was used to quantify the recovered product. Finally, all of the purified amplicons were pooled for paired-end sequencing on an Illumina Miseq PE300 platform (Shanghai Majorbio Medical Technology Co., Ltd., Shanghai, China).

### 2.4. Sequence Data Processing and Analysis

Sequencing of the 16S rDNA amplicon was performed using an Illumina Miseq platform (Illumina, San Diego, CA, USA) at MAJORBIO, Shanghai, China. The raw 16S rDNA gene sequencing reads were quality-filtered with fastp [30], and FLASH (v1.2.7) [31] was used to assemble the paired-end reads as follows: (1) filter the bases with a tail mass value of less than 20 and set a 50 bp window. If the average quality value in the window is less than 20, cut back bases from the window, filter the reads of less than 50 bp after quality control, and remove ‘N’ base reads; (2) consider a minimum overlap of 10 base pairs (bp) and allow a maximum mismatch ratio in the overlap area of 0.2; (3) perform exact barcode matching; distinguish the samples according to the barcodes and primes at the beginning and at the end of the sequence and adjust the sequence direction. Clean reads were analyzed using the QIIME2 [32] software package (v2020.6). The QIIME2 tool DADA2 [33] was used to denoise the optimized sequence after quality control splicing (default parameters), yielding amplicon sequence variants (ASVs). Subsequently, the 16S SILVA reference database classifier (v138) was used to classify ASVs with a threshold of 70% sequence similarity. Microbial diversity and community composition were analyzed using the vegan package in R (v3.5.1). Alpha diversity indices including Ace, Chao1, Shannon, and Simpson were applied in analyzing the complexity of species diversity in the samples. Principal component analysis (PCA) plots were constructed with the vegan package in R (v3.3.1) [34]. Alpha diversity and Beta diversity were calculated by the QIIME2 script. Histograms and correlations of the bacterial taxa were obtained for different time periods by correlation analysis using the ggplot2 and heatmap packages in R (v3.3.1). Functional predictions were performed using the PICRUSt2 software (v2.2.0-b) [35].

### 2.5. Quantification of Bacterial Communities

The DNA extracted from cotton aphid ovaries at 1 d and 3 d after parasitization was quantified with ‘absolute’ real-time qPCR, and unparasitized cotton aphid ovaries from the same period were used as controls. The copy numbers of target genes (Appendix A) were calculated from a standard curve based on serial-dilution gradients of the target sequences cloned in the pEASY-T1 cloning vector (TransGen Biotech, Beijing, China). qPCR reactions were performed on the StepOnePlus Real-Time PCR system (Applied Biosystems, Foster City, CA, USA) using a 20 µL reaction mixture containing 10 µL of 2 × TransStart Green qPCR SuperMix (TransGen Biotech), 0.8 µL of each 10 mM primers, 0.4 µL of 50 × ROX, 1 µL of template DNA, and 7.4 µL of H_2_O. The cycling conditions used were 95 °C for 2 min, 40 cycles of 95 °C for 5 s, and 60 °C for 30 s, using the corresponding standard curve in each reaction. Each sample was replicated three times. Copy number differences between samples were assessed by SPSS 20.0, Mann–Whitney U test (group = 2), and Kruskal–Wallis test (n > 2).

## 3. Results

### 3.1. Analysis of the 16S rDNA Sequencing Results

After quality filtering and removal of chimeric sequences using DADA2 on the QIIME2 platform [32], all groups yielded a total of 1,074,148 reads, with an average of 33,567 reads per group. In total, 1533 amplicon sequencing variants (ASVs) were identified in 24 individuals; the average ASV number in each sample ranged from 18 to 89, and the average ASV length in each sample was 426 bp. The sample rarefaction curve (S1A) and the Shannon index rarefaction curve (S1B) showed that the number of sequenced ASVs was sufficient and that increasing the sample volume would not produce more ASVs. Similarly, Good′s coverage estimates showed that the sequencing depth captured most bacterial species of the microbiota in cotton aphid ovaries.

### 3.2. Community Diversity Analyses

Alpha diversity and bacterial composition were investigated for the ovaries of cotton aphids parasitized for 1 day and 3 days during the third-instar nymph and adult periods, with unparasitized *A. gossypii* as controls. As shown in Figure 1, the Ace index, Chao index, and Shannon index of the control group’s ovaries were greater than those of the ovaries of parasitized third-instar nymph and adult *A. gossypii* on day1. However, the Ace index, Chao index, and Shannon index of the control group were lower than those of adults parasitized for 3 days. The statistical analyses revealed that parasitism decreased the richness and diversity of the ovarian communities when nymphs and adults were parasitized for 1 day but increased their richness and diversity when adults were parasitized for 3 days.

PCA (Principal Component Analysis) was used to analyze the similarities between the ovarian microbiota in different samples. The PCA used ANOSIM [36] to analyze the ASV data of parasitized and unparasitized cotton aphids and to display the distances and gaps between samples in a two-dimensional coordinate diagram. The PCA analysis of the cotton aphid ovary microbiomes in different periods revealed that the community composition of cotton aphids in three treatment groups was obviously separated by *B. communis*, indicating that the microbial community in the ovaries of cotton aphids was changed by *B. communis* parasitization (Figure 2).

### 3.3. Analysis of the Microbial Community Composition in th Ovaries

After removing chloroplast and mitochondrial ASVs, the remaining data for the ovaries of cotton aphids were analyzed at a taxonomic level. A total of 1533 ASVs were identified with 99% sequence similarity. The 1533 ASVs were categorized into 15 phyla, 26 classes, 60 orders, 96 families, and 170 genera. The dominant phyla were *Proteobacteria*, *Actinobacteriota*, and *Bacteroidota*, the most abundant of which was *Proteobacteria*. At the genus level, *Buchnera*, *Achromobacter*, *Rhodococcus, Arsenophonus*, *Serratia*, *Ochrobactrum, Sphingobacterium*, *Rahnella1*, *Stenotrophomonas*, and *Pseudomonas* were the 10 most abundant genera (Figure 3). The community composition analysis showed that the *A. gossypii* ovaries microbiota was dominated by the endosymbiotic bacterium *Buchnera*. The relative abundance of *Serratia*, *Microbacterium*, *Leucobacter*, *Achromobacter*, and *Rhodococcus* in the adult stage was significantly higher than in the third-instar nymph stage.

### 3.4. Microbial Community Alterations Due to Parasitism

In the third-instar nymph stage, the relative abundance of *Buchnera* increased slightly at 1 d after parasitization (92.8%) compared to the control (92.7%). However, after three days, parasitism (82.9%) decreased the relative abundance of *Buchnera* in the ovaries compared to the control (90.56%). Likewise, the relative abundance of *Buchnera* (70.36%) increased in the adults 1 d after parasitization compared to the control (36.09%). However, after parasitization for 3 d, the relative abundance of *Buchnera* (37.11%) was lower than in the control (52.01%). The relative abundance of *Arsenophonus* increased in the third-instar nymph stage 1 d after parasitization (6.24%) compared to unparasitized ovaries (5.17%), while it decreased 3 d after parasitization (3.26%) compared to the control ovaries (8.56%). Similarly, the change in abundance of the *Arsenophonus* in the adults was the same as in the third-instar nymphs. Moreover, the relative abundance of *Enterobacteriaceae* in the third-instar nymphs 1 d after parasitization (0.09%) was reduced compared to the control (0.9%). However, the relative abundance of *Enterobacteriaceae* increased in the third-instar nymphs 3 d after parasitization (1.29%) compared to the control (0.25%). In third-instar nymphs that were parasitized for 3 d, the relative abundance of a variety of facultative symbionts increased, including *Ochrobactrum* (1.1%), *Sphingobacterium* (0.16%), *Rahnella1* (5.26%), *Stenotrophomonas* (0.15%), and *Pantoea* (1.6%). In the adult stage, the relative abundance of *Serratia* decreased 1 d after parasitization (from 5.3% to 1.5%) but increased 3 d after parasitization (from 3.3% to 6.8%). Meanwhile, parasitism increased the relative abundance of *Pseudomonas* and *Pantoea*. Furthermore, parasitism reduced the relative abundance of *Rhodococcus*, with an obvious lower relative abundance of *Rhodococcus* (5.7%) 1 d after parasitization compared to the control (19.03%) (Figure 3). In order to more clearly show the dynamics of the microbiota in the ovaries of cotton aphids after parasitization, the 20 most abundant genera are shown in a relative-abundance clustering heat map (Figure 4). Clustering was based on the abundance of species, and all four replicates of each treatment were aggregated into a single branch.

### 3.5. Functional Prediction

To fully understand the changes in the microbial community in *A. gossypii* ovaries after parasitization by *B. communis*, the functional genes of 16S rDNA samples were predicted with Picrust2 software and compared with the Cluster of Orthologous Groups (COG) database. The majority of functional prediction categories in the ovary microbiome were related to cells and metabolism. The top five were “amino acid transport and metabolism,” “inorganic ion transport and metabolism,” “energy production and conversion,” “transcription,” and “cell wall/membrane/envelopment biogenesis.” Additionally, we also investigated “defense mechanisms” and “replication, recombination, and repair” (Figure 5).

### 3.6. RT-qPCR of Core Symbiotic Bacteria

In order to verify the abundance of core symbiotic bacteria, the copy numbers of *Buchnera*, *Arsenophonus* and *Serratia* in each treatment group were determined by qPCR. In the third-instar nymph stage, the copy numbers of *Buchnera* increased 1 d after parasitization but decreased significantly 3 d after parasitization. The changes of *Arsenophonus* in the adult stage were the same as those of *Buchnera* (Figure 6A). The copy numbers of *Arsenophonus* were also further verified with qPCR and varied equally in adults and third-instar nymphs, with an increase in relative abundance 1 d after parasitization, but a significant decrease 3 d after parasitization (Figure 6B). *Serratia* was verified in adult aphid ovaries, and the copy numbers of *Serratia* appeared decreased in adults 1 d after parasitization but increased in the same 3 d after parasitization compared to the control (Figure 6C).

## 4. Discussion

In this study, the bacterial community dynamics in *A. gossypii* ovaries were investigated with high-throughput 16S rDNA amplicon sequencing. The effects of parasitism by *B. communis* on the *A. gossypii* ovarian microbiome composition were investigated. A total of 1533 ASVs were identified including 15 phyla, 26 classes, 60 orders, 96 families, and 170 genera. The cotton aphid ovarian microbiome was mainly composed of the obligate symbiont *Buchnera* and a variety of facultative symbionts such as *Arsenophonus, Achromobacter*, and *Serratia*. The relative abundance of *Buchnera* in the aphid ovary increased when third-instar nymphs were parasitized for 1 d but decreased 3 d after parasitization. In the adult stage, the relative abundance of *Buchnera* also increased when the adults were parasitized for 1 d and decreased when they were parasitized for 3 d. The changes in the relative abundance of *Arsenophonus* at each stage were similar to those of *Buchnera.* In addition, the relative abundance of *Serratia* remarkably decreased in the adults 1 d after parasitization and increased in the same 3 d after parasitization. As far as we know, this is the first time that *Rahnella1*, *Pantoea*, *Ochrobactrum*, and *Phyllobacterium* were found in *A. gossypii* ovaries. Finally, RT-qPCR was performed on *Buchnera, Arsenophonus*, and *Serratia* to further verify their relative abundance. The results of RT-qPCR were the same as the results of 16S rDNA sequencing. Our results provide a strong foundation for insect ovary microbiome research. Furthermore, this is the first report on *A. gossypii* ovarian symbiotic bacteria and changes in the ovarian microbial community after parasitization.

Aphids feed exclusively on the phloem sap of plants. Most aphids contain *Buchnera* symbionts, which provide aphids with the necessary amino acids [17]. In this experiment, *Buchnera* was also the main genus found in the bacterial community of the *A. gossypii* ovary. Facultative symbiotic bacteria were also found, including *Achromobacter*, *Rhodococcus*, *Arsenophonus*, *Serratia*, *Pseudomonas*, *Rahnella1*, *Pantoea*, *Ochrobactrum*, and *Phyllobacterium*. Previous studies have shown that the facultative symbionts which can infect aphids are mainly *Arsenophonus*, *Serratia*, *Wolbachia*, *Regiella*, *Rickettsia*, *Pseudomonas*, and *Hamiltonella* [37,38,39,40,41,42]. However, in this study, *Wolbachia*, *Hamiltonella*, *Rickettsia*, and *Regiella* were not found. All of the samples in this study contained *Arsenophonus*; however, the relative abundance of *Serratia* over 1% only in the adult stage, and the relative abundance of these two facultative symbiont bacteria was very low, which may be related to the development stage of the aphids.

In host–parasitoid interactions, symbiotic bacteria play an important role. Previous research has shown that parasitism can alter the host microbial community and reduce host fecundity [28]. In our results, we found that after *A. gossypii* is successfully parasitized, the primary symbiotic bacteria *Buchnera* in the ovaries decrease in third-instar nymphs and adults 3 d after parasitization. *Buchnera* lives primarily in special aphid cells called “bacteriocytes,” which increase in number and size during nymph development [43]. It provides the nutritional requirements and energy for the growth and molting of the worm during its development to adulthood [44]. Pers et al. showed that early in nymphal development, *Buchnera* is active in the metabolism of essential amino acids and vitamin B [45]. It is clear that amino acids are not only incorporated into proteins in insects but also broken down into metabolic intermediates, such as neurotransmitters and nucleotides, as well as to provide energy [46]. In previous studies, it was shown that the relative abundance of *Buchnera* in *A. gossypii* increased after parasitization and the relative abundance of *Buchnera* was higher in nymphs than in adults [29]. In this paper, we confirmed the previous conclusions and also performed RT-qPCR to validate our results. However, in contrast to the previous results, the relative abundance of *Buchnera* was temporarily elevated 1 d after parasitization, and thus it is possible that *Buchnera* provides nutrition for the growth of *A. gossypii* or the development of *B. communis* eggs. However, the relative abundance of *Buchnera* decreased in adult 3 d after parasitization, and because *Buchnera* is the primary symbiotic bacterium that provides nutrients, we speculate that the reduction in *Buchnera* may contribute to the reduced fecundity of *A. gossypii*, consistent with the shift to “amino acid transport and metabolism” from our functional prediction.

In host–parasitoid interactions, facultative symbiotic bacteria also play an important role in the defense against parasitoid wasps. We also found *Arsenophonus* and *Serratia* in the microbial community of *A. gossypii* ovaries. The facultative symbiotic bacteria *Arsenophonus* plays an important role in insect hosts, regulating reproduction by killing offspring and increasing the dietary breadth of its hosts [47]. Tian et al. showed that *Buchnera* relative abundance increased when cotton aphids were infected with *Arsenophonus* [47]. A typical defensive symbiont, *Hamiltonella defensa*, protects aphids from parasites by disrupting wasp development. A successful defense requires that *H. defensa* infect the phages APSEs (Acyrthosiphone pisum secondary endosymbiont), which play other key roles in maintaining mutualism [48]. Boyd B.M. et al. also concluded that APSE bacteriophages are most closely associated with *Arsenophonus* and *Morganellaceae* through genomic and phylogenetic analyses [49]. In the results shown here, after parasitization, all samples contained *Arsenophonus*, and the changes in *Arsenophonus* were same as those in *Buchnera. Arsenophonus* increased in third-instar nymphs and adults 1 d after parasitization and decreased in third-instar nymphs and adults 3 d after parasitization. The results of the changes in the relative abundance of *Arsenophonus* were not the same as those for the *L. japonica* parasitism of *A. gossypii* [29]. Therefore, we hypothesize that the nutrients provided by *Buchnera* are required by *Arsenophonus* and that the reduction in *Arsenophonus* may also be the reason for the decrease in *A. gossypii* eggs. *Serratia* is a common facultative symbiotic bacterium in *A. gossypii* which improves host adaptability and heat tolerance [50]. Zhou et al. found that in pea aphids, *Serratia* can promote aphid host growth and development by enhancing fatty acid biosynthesis [51]. Moreover, *Serratia* and *Rickettsia* could suppress the population density of *Buchnera* [52]. Recent studies have shown that the presence of *Serratia* in *Acyrtosiphum pisum* can influence the development and fitness of parasitoid wasps *Aphidius ervi* offspring and affect their foraging strategy [53]. In this study, *Serratia* abundance changed in the adult stage, and its relative abundance increased after parasitization, indicating that *Serratia* may be involved in resisting the parasitoid effect of *B. communis* and ensuring the growth of the host aphids, which may be related to the functional predicted categories “amino acid transport and metabolism” and “energy production and conversion.” The copy numbers of these two facultative symbiotic bacteria and the observed changes after parasitization were further verified and were consistent with these results.

## 5. Conclusions

In conclusion, we described for the first time dynamic changes in the structure and composition of the ovarian microbial community over time and alterations of the microbiota within aphid ovaries as a result of parasitism. We found the community consisted of the obligate symbiont *Buchnera* and facultative symbionts including *Achromobacter*, *Rhodococcus*, *Arsenophonus*, *Serratia*, and *Pseudomonas*. The relative abundance of *Buchnera* in the aphid ovary increased in both third-instar nymphs and adults 1 d after parasitization but decreased 3 d after parasitization. The changes in the relative abundance of *Arsenophonus* at each stage were similar to those observed for *Buchnera.* In addition, the relative abundance of *Serratia* remarkably decreased in the adults 1 d after parasitization and increased in the same 3 d after parasitization. Moreover, RT-qPCR was performed on *Buchnera*, *Arsenophonus*, and *Serratia* to verify these results. This research suggests that these alterations in symbiotic bacteria may play an important role in the regulation of host reproduction by parasitoid wasps and thus may act as a potential driving force in complex parasitoid–host interactions. These results provide a framework for further investigation of the effects of parasitoid wasps on host ovaries and of the molecular mechanisms of host reproduction regulation by parasitoid wasps.

## Figures and Tables

**Figure 1 insects-14-00314-f001:**
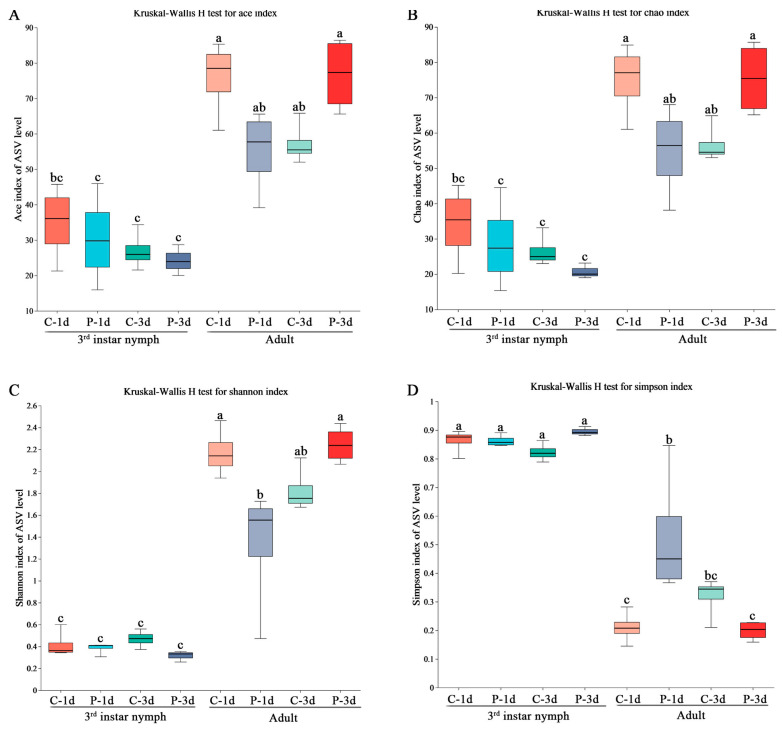
Box diagram of (**A**) Ace, (**B**) Chao, (**C**) Simpson, and (**D**) Shannon Indices of symbiotic bacterial communities in the ovaries of *A. gossypii* at different ages and stages. The treatment groups on the x-axis are abbreviated as follows: C-1d = ovaries of unparasitized aphids 1 d, P-1d = ovaries of parasitized aphids, 1 d, C-3d = ovaries of unparasitized aphids, 3 days, P-3d = ovaries of parasitized aphids, 3 days. Different superscript letters (a, b, c) represent statistically significant differences among groups (*p* < 0.05).

**Figure 2 insects-14-00314-f002:**
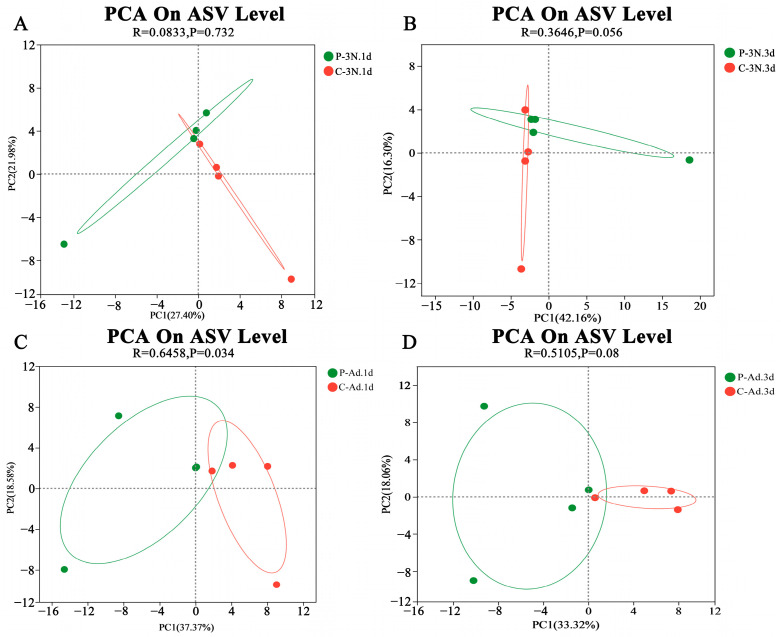
Principal Component Analysis (PCA) based on ANOSIM distances between treatments. (**A**,**B**) PCA of ovaries of third-instar nymphs that parasitized and unparasitized for 1 d and 3 d, (**C**,**D**) PCA of ovaries of adults that parasitized and unparasitized for 1 d and 3 d. The treatment groups are abbreviated as follows: C-3N.1 d = ovaries of third-instar nymphs that unparasitized for 1 d, P-3N.1 d = ovaries of third-instar nymphs that were parasitized for 1 d, C-3N.3 d = ovaries of third-instar nymphs that unparasitized for 3 d, P-3N.3 d = ovaries of third-instar nymphs that were parasitized for 3 d, C-Ad.1 d = ovaries of adults that unparasitized for 1 d, P-Ad.1 d = ovaries of adults that were parasitized for 1 d,C-Ad.3 d = ovaries of adults that unparasitized for 3 d, P-Ad.3 d = ovaries of adults that were parasitized for 3 d. ASV means amplicon sequencing variants.

**Figure 3 insects-14-00314-f003:**
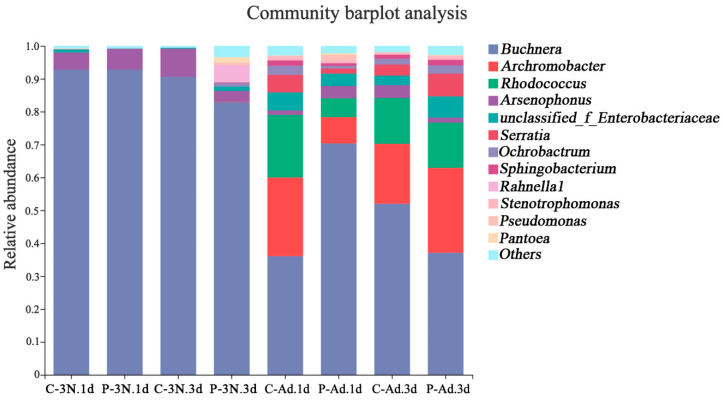
Bacterial community composition plots in parasitized and unparasitized *A. gossypii* ovaries. Bar chart showing the relative abundance of dominant bacterial genera (over 1%).

**Figure 4 insects-14-00314-f004:**
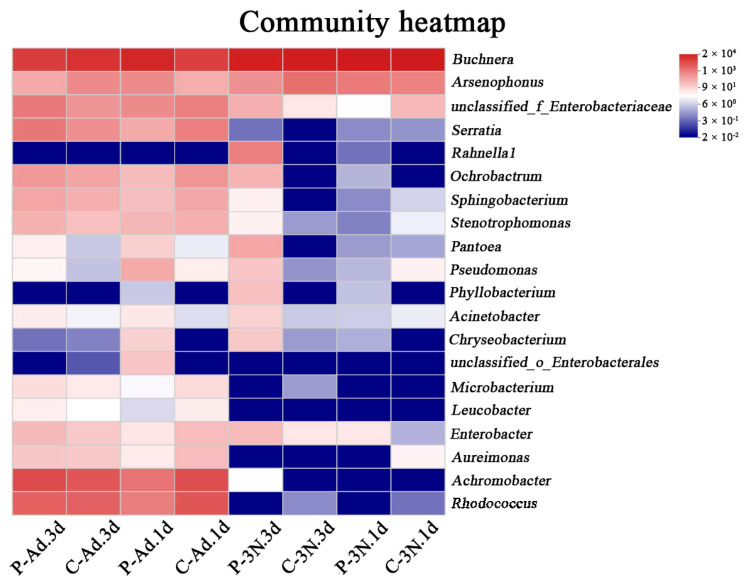
A clustered heat map of the 20 most abundant genera in the bacterial community. The columns represent the different treatments, and the rows represent bacterial genera assigned with ASVs (amplicon sequencing variants). Hierarchical clustering analysis trees of taxonomic genera are shown on the right.

**Figure 5 insects-14-00314-f005:**
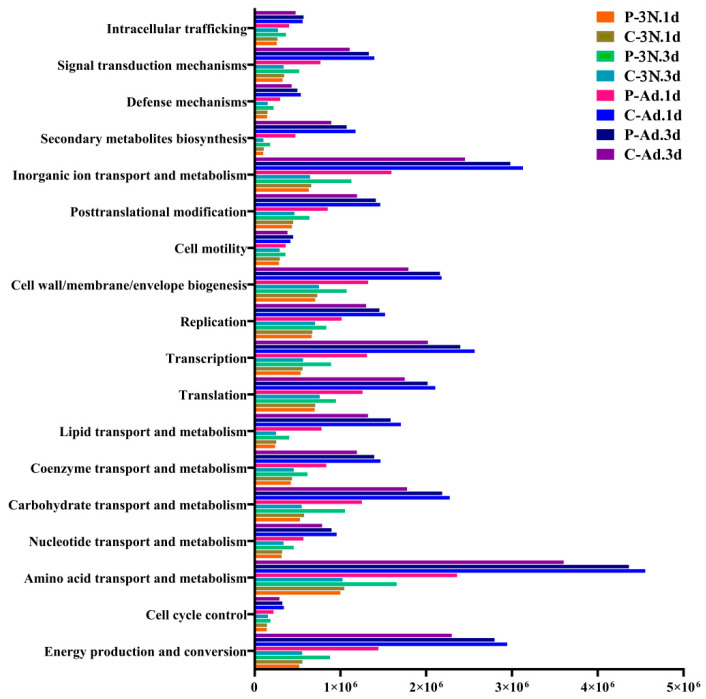
Comparison of COG (Cluster of Orthologous Groups) function prediction of microorganisms in ovaries at different developmental stages and times after treatment.

**Figure 6 insects-14-00314-f006:**
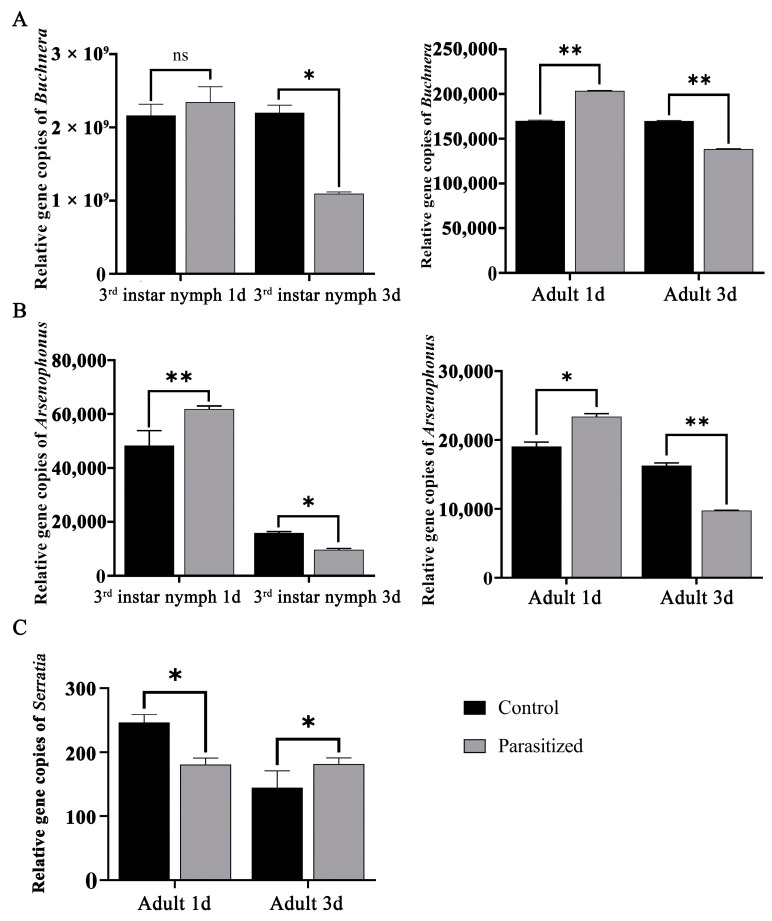
(**A**) Copy numbers of *Buchnera* (**A**), *Arsenophonus* (**B**), and *Serratia* (**C**), 16S rDNA gene in the ovaries of parasitized and non-parasitized cotton aphids. We did not perform RT-qPCR for nymph stage *Serratia* due to its low relative abundance. Asterisks define the *p*-values as follow: * < 0.05 and ** < 0.01, ns indicates that no significant difference in copy number.

## Data Availability

The datasets used and analyzed during the current study can be supplied by the corresponding author upon reasonable request.

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
