# Peer review of "Parasitization of Aphis gossypii Glover by Binodoxys communis Gahan Causes Shifts in the Ovarian Bacterial Microbiota"

_insects, 2023, doi:10.3390/insects14040314_

Round 1

Reviewer 1 Report

The draft of article number 2246520 submitted to the Insects, MDPI, entitled “Parasitization of Aphis gossypii Glover by Binodoxys communis Gahan causes shifts in ovarian bacterial microbiota” carried interesting results in the text that needs some revision for the improvement of the draft. Some suggested changes for example are in the comments portion to revise and improve the manuscript. Please find suggested corrections, reference writing, journal-style format, author’s instructions, use of abbreviations and missing information for revision. 

 Line 31-32: After parasitization, the relative abundance of Buchnera in the aphid ovary increased at the stage of third-instar nymph 1 d, but decreased at third-instar nymph 3 d” Please rephrase the sentence and explain the third-instar nymph 1 d and third-instar nymph 3 d

Abstract

Abstract formatting may be according to the author’s instructions

Line 67: Buchnera” Pl use complete scientific names of bacteria

“at the stage of third-instar nymph 1 d, but decreased at third-instar nymph 3 d” Please rephrase the sentence and explain the 1d and 3d

Figure 2: what is ASV level” please explain the abbreviation

Figure 6: COG” please explain

Figure 7: please write the complete set of information “3rd nymph 1d” 3rd nymph 3d” 3rd instar nymph 1d

Conclusion: portion is too long please write precisely.

Please double-check for inconsistencies in Journal style/formatting/ authors instructions, double spaces, spellings of the words, English vocabulary, missing italics, scientific names, excessive/missing information, etc.

Author Response

Dear reviewers, We are grateful to the reviewers for the constructive comments on our manuscript "Parasitization of Aphis gossypii Glover by Binodoxys communis Gahan causes shifts in ovarian bacterial microbiota". They have enabled us to revise and greatly improve our manuscript. Our responses to reviewer comments are listed below in point-by-point fashion. We hope that your will agree that the manuscript is now suitable for publication in Insects and look forward to hearing from you at your earliest convenience. Sincerely yours, Dr. Jin-Jie Cui

Reviewer 2 Report

The manuscript submitted by Jinming Li, et al. entitled: “Parasitization of Aphis gossypii Glover by Binodoxys communis Gahan causes shifts in ovarian bacterial microbiota“ is a study of the possible influences of a parasitoid wasp on the bacterial community composition on two life stages of aphids.   The authors present an experiment that has sufficient sample sizes and replication and they follow standard methods of microbiome analysis.

There are some minor problems with this study:  lack of clarity on some points, missing methods, and missing Supplementary materials (it is not clear which they are since all that is in the text is “S1A” and “S1B”, but there is also a reference to a “Table S1”). It is therefore impossible to evaluate this part of the manuscript.  

General comments: You have multiple references to larval (versus nymphal) aphids, which should be corrected and are listed below. This is a complex system of a hemimetabolous host and a holometabolous parasitoid, so references to “larva” for both host and parasitoid is confusing and incorrect.  Further, there are inconsistencies in this designation, using “larva” or “L” in some places, while also referring to it as “nymph” or “nymphal” in others. This needs to be corrected.  

There is no demonstration of an analysis to determine whether there is an interaction between age (days post-parasitism, dpp) and parasitization status. In both nymphs and adults post-parasitoid exposure, it appears that some bacterial taxa are reduced from 1 dpp versus 3 dpp. This could be due to the effect of the parasitoid, except this is also observed in the controls. How might you address the potential effects of post-molting microbial stabilization or fluctuations due to age? Aphids do not have a long life span, so fluctuations in microbial communities from day to day or between molts may be normal.

Specific comments:

L18: “16s” should be “16S”

L26: …the effects on the symbiotic bacteria …”, the effects of parasitism on the symbiotic bacteria? 

L31-33: Could this increase and decrease from days 1 and 3, respectively post-parasitism represent fluctuations/stabilization of microbiota that may occur after a molt? it was observed in both parasitized and control organisms.

L34-36: Again, could this fluctuation of Serratia abundance reflect a stabilization of the bacterial taxon as the aphid developed? 

L38-39: “RT-qPCR analysis was performed…” This is a method, not a result. What was the result (summary)?

L69: “Serratia symbiotica and Hamiltonella defense protect aphids against heat shock and provide resistance to parasitoid wasps”.Did you confirm which Serratia species you detected? 

L80-2: “Female parasitoid wasps use their ovipositors to lay eggs inside aphid larvae and their larvae…” [presumably the wasp?] “…eventually cause the death of the host by consuming the living host’s developing tissue.” Apart from the confusion about which immature stage is being discussed, the use of “aphid larva” is not correct. 

L95: same comment re: “larvae”

L110-113: In the controls (i.e. without parasitoid eggs), did you do a mock injection or piercing to simulate oviposition? Were aphids injected with beads to simulate the introduction of a foreign body (and subsequent triggering of immune defenses)? 

L113-120: Sample Collection: Were the aphids surface-sterilized before dissection or just after dissection? Was the carcass then used for the DNA extraction or were the ovaries removed and used for DNA extraction? If the latter, why would the removal of wasp eggs have made a difference in DNA purity?  

Though perhaps outside the scope of this study, it would be interesting to know if the viability and/or microbiome of the wasp embryos were affected by the aphid hosts’ microbiota.

L161: Table S1. Where is this? 

L176-7: “the average ASV length in each sample ranged from 18 bp to 89 bp “ This seems short. Even with MiSeq, the range of read lengths is usually between 75-150 bp after trimming and quality control. 

L178:S1A and S1B. Where is this?

L 180-181: Good’s coverage estimates (what are these data?)

L182 Community diversity analysis (should be “analyses”, since there is more than one)

L 182-92: You do not have any statistics comparing the difference in days post parasitism (DPP), but you are comparing the effects between days 1 and 3, and between life stages (nymph versus adults).  Did you look to see if you had an interaction and what if any effect these had on your conclusions?

186-186: why were these alpha diversity indices not described in the methods?

Figure 1: While the use of “L” for 3rd instar nymphs has been defined, it would be better to change this in the figures. This should just be a matter of changing the labels in the mapping file and re-running this through R. Kruskal Wallis is a between-group analysis, but you have no stats shown. What were the p-values/significance (* vs NS)? This can be added to the figure in R. Did you use pairwise comparisons? 

Figure 1: Which Chao index and which Simpson index was used? The interpretations may be different.

L191: same comment re: larva

Figure 2: The resolution of Figure 2 was very hard to read. If I am reading it correctly, the PCA for the L3.1d was not significant, nor was the R square value very high. I think it’s R, but I can’t see it. 

Figures 3 and 4: I’m not sure why you need Figure 3, since you could briefly state the four dominant taxa in your text.  Figure 4 doesn’t need to be redone, but I would suggest that for ease of comparison, you could plot it as a split plot, with controls on top and parasitized aphid stages on the bottom, so you can better visually compare this with Figure 2. 

Page 8:  This entire page you are describing pairwise comparisons for each of your core taxa, basically re-describing your figures. 

L274-5: Why is this interesting?

L339-41: You observed Serratia in Adults, but not nymphs. Why might that be?

L354-5: Figure 7: it’s not necessary to include a barplot, but perhaps in the legend, you could indicate that Serratia was not detected by qPCR (or not tested?) in nymphs. You should include what statistic you used to do your pairwise comparisons.

L367: “In addition, the relative abundance of Serratia remarkably decreased at 1 d after parasitization and increased at 3 d after parasitization. “ To which analysis are you referring? The sequencing data or the qPCR? I don’t know if you can say that it decreased if you don’t have any data before this (e.g. day 0 or before exposure to parasitism). If you are referring to the qPCR data, you have no nymphal data for comparison for Serratia. 

L378: “Faculative” should be “Facultative”

L382: “Rickettisa” should be “Rickettsia”

L383-35: “…only adults contained Serratia, which may be related to the development time of aphids”, yet in your sequencing data you presented low levels of Serratia in your heat map (Figure 5). 

L389-91: To what are you referring as a “larval worm”? In parasitology, a “larval worm” might be an immature stage of a helminth, but some insects (and even some vertebrates) have been referred to as “worms” as part of their common names. If you are going to use “worm” in this context, it should be defined early on in the text, but its use is not strictly correct. 

L390 “bacterial cells” <—Do you mean “bacteriocytes”? Has this terminology changed? 

L392: “worm” <—same comment about larval. 

L393–4: Pers and Hansen do not refer to a worm period

L399: same comment re: larva

L411-413: Arsenophonus is involved in manipulating reproduction, but is it known to do this in Binodoxys communis? it is not thought known to protect against parasitoids or fungi (Wulff et al 2013)

L415-417: What is this sentence about? Is there something missing? What do the APSE bacteriophages have to do with this? Are you suggesting that a bacteriophage from a bacterium related to Arsenophonus also infects and influences Buchnera?! You didn’t test this! 

L421-423: While the Discussion is where speculation is acceptable, you did not examine the egg-laying capacity of the aphids post-parasitism because you sacrificed them for the study (Unless you did, in which case you did not mention this in the methods).  

L429-31: “Serratia abundance changed in the adult stage and relative abundance increased after parasitization, indicating that Serratia may be involved in resisting the parasitoid effect of B. communis …” It’s difficult to see in the heat map what exactly was happening in the adults. It looks like control adults 1dpp and parasitized adults 3dpp were very similar in Serratia abundance; while in nymphs both the controls and parasitized individuals showed a relative increase in abundance from days 1 to days 3 (in other words, they have similar trends). In contrast, your Figure 4 (relative abundance) seems to show no Serratia in the nymphs, no change in control adults from d1 versus d3, and an increase in parasitized adults from d1 to d3.  

L429-31: if Serratia is known to affect parasitoid fitness, it would have been a good idea to have checked the parasitoids for viability.

L449-450: Same comment as in L31-36

L 451-2: It would not be far-fetched to assume parasitism would affect the reproductive rate of aphids, so it is difficult to see how you would tease apart the effects that each would play (symbionts versus parasitoids). Complete curing and then stepwise addition of key symbionts with and without parasitism might work, but the removal of Buchnera might result in the nonviability of the aphids. 

Author Response

(The authors gave the same response as above.)
